# Technical note: Conservative storage of water vapour - practical *in-situ* sampling of stable isotopes in tree stems

Ruth-Kristina Magh[1,2], Benjamin Gralher[3], Barbara Herbstritt[3], Angelika Kübert[4,5], Hyungwoo Lim[1], Tomas Lundmark[1], John Marshall[1]

[1]Forest Ecology and Management, Swedish University of Agricultural Sciences, Umeå, 90183, Sweden
[2]Terrestrial Ecohydrology, Friedrich Schiller University, Jena, 07749, Germany
[3]Hydrology, Albert Ludwigs University, Freiburg, 79098, Germany
[4]Ecosystem Physiology, Albert Ludwigs University, Freiburg, 79103, Germany
[5]Micrometeorology, University of Helsinki, 00100 Helsinki, Finland

*Correspondence to*: Ruth-Kristina Magh (ruth.magh@posteo.net)

**Abstract.** Using water stable isotopes to track plant water uptake or soil water processes has become an invaluable tool in ecohydrology and physiological ecology. Recent studies have shown that laser absorption spectroscopy can measure equilibrated water vapour well enough to support inference of liquid stable isotope composition of plant or soil water, on-site and in real-time. However, current *in-situ* systems require the presence of an instrument in the field. Here we tested, first in the lab and then in the field, a method for equilibrating, collecting, storing, and finally analysing water vapour for its isotopic composition that does not require an instrument in the field. We developed a vapour storage vial system (VSVS) that relies on *in-situ* sampling into crimp neck vials with a double-coated cap using a pump and a flow meter powered through a small battery and measuring the samples in a laboratory. All components are inexpensive and commercially available. We tested the system's ability to store the isotopic composition of its contents by sampling a range of water vapour of known isotopic compositions (from -95 to +1700‰ for $\delta^2H$) and measuring the isotopic composition after different storage periods. Samples for the field trial were taken in a boreal forest in northern Sweden. The isotopic composition was maintained to within 0.6 to 4.4‰ for $\delta^2H$ and 0.6 to 0.8‰ for $\delta^{18}O$ for natural-abundance samples. Although $^2H$-enriched samples showed higher uncertainty, they were sufficient to quantify label amounts. We detected a small change in the isotopic composition of the sample after long storage period, but it was correctable by linear regression models. We observed the same trend for the samples obtained in the field trial for $\delta^{18}O$ but observed higher variation in $\delta^2H$ compared to the lab trial. Our method combines the best of two worlds, sampling many trees *in-situ* while measuring at high precision in the laboratory. This provides the ecohydrology community a tool that is not only cost-efficient but also easy to use.

## 1    Introduction

Since the introduction of isotope-ratio infrared spectrometers (IRIS), the analysis of water stable isotope samples has become much more popular in many fields, e.g., in hydrogeologic, watershed, oceanographic or eco(hydro)logical studies (Tweed et

al., 2019; Oerter and Bowen, 2017; Oerter et al., 2019; Beyer et al., 2020; Quade et al., 2019; Volkmann and Weiler, 2014; Volkmann et al., 2016b). This has led to an increased utility of water stable isotopes also in applications, where the interest of inferring plant water uptake depths/patterns and water movements through the soil matrix has grown tremendously

(Eggemeyer et al., 2008; Liu et al., 2010; Beyer et al., 2016; Magh et al., 2020).

Until recently, however, samples of soil matrix- or plant tissue-bound water needed to be obtained destructively to extract the water samples. A method that is frequently used is cryogenic vacuum extraction, where a sample undergoes heating under vacuum, with the bound water evaporating in the process and subsequently being captured in a cryogenic trap (Ingraham and Shadel, 1992; Koeniger et al., 2011; Orlowski et al., 2013, 2016). The method was preferred because the

assumed completeness of the water extraction was thought to eliminate fractionation. However, it has recently been heavily criticised for introducing biases due to artefacts coming from an exchangeable organic hydrogen pool in the plant biomass (Chen et al., 2020; Allen and Kirchner, 2022) and representing mainly the tightly bound water in the soil (Orlowski et al., 2016; Zhao et al., 2013) .

A recently developed method based on direct vapour equilibration reduces the co-extraction of organic compounds and

increases sample throughput (Millar et al., 2018; Wassenaar et al., 2008). One of the biggest advantages of *in-situ* equilibration techniques is that water from plants and soils can be sampled at high temporal resolution without altering their physiology or physical properties (Kühnhammer et al., 2021). This is particularly noticeable when repeatedly sampling the same tree for cores, as water transport repeatedly gets disrupted, while when using the in-situ approach this only happens once. In the soil, the recurrence of drilling eventually alters water flow of the entire plot since it opens many preferential

flow channels in the same vicinity. Therefore, *in-situ* measurements of water stable isotopes have gained popularity and have been proposed a way forward to disentangle isotopic processes in the critical zone or the soil-vegetation-atmosphere continuum (Rothfuss and Javaux, 2017; Beyer et al., 2020).

*In-situ* measurement systems are based on direct inferences of liquid water isotopic composition from equilibrated water vapour from the soil or the plant (for a detailed review see Beyer et al., (2020)). The vapour is collected either using a gas-

permeable membrane (the utility of which was proven by Herbstritt et al., (2012) buried in the soil (Rothfuss et al., 2013; Volkmann et al., 2016b; Volkmann and Weiler, 2014; Kübert et al., 2020) or in the xylem of woody species (Volkmann et al., 2016a, b; Seeger and Weiler, 2021), or drawing equilibrated water vapour from a borehole in the xylem directly (Marshall et al., 2020; Kühnhammer et al., 2021). Additionally, it is possible to measure the isotopic composition of plant transpiration and evapotranspiration *in-situ*, using gas exchange chambers in the lab (Simonin et al., 2013; Dubbert et al.,

2017), as well as in the field (Kübert et al., 2019; Dubbert et al., 2013; Wang et al., 2013).

The biggest advantage of these *in-situ* systems is their ability to monitor real-time changes in water uptake and subsequent transport in plants and/or in soils and produce immediate data. The biggest disadvantage is the need for an IRIS at the site of measurement, which requires shelter, protection against vandals, and most importantly, access to a continuous power source. Additionally, the *in-situ* setup in practice is limited in spatial resolution, as it requires tubing at the length of the distance

from the sampling place to the IRIS, which is advisably kept short as increased tubing length increases the possibility of

condensation (Beyer et al., 2020; Kühnhammer et al., 2021). These factors limit the utility of *in-situ* measurement systems to field sites in vicinity to civil infrastructure, which potentially leads to research sites chosen because of proximity to power rather than suitability as research location, and therefore, location biases (e.g. monitoring wildlife in vicinity to universities (Piccolo et al., 2020), or the location of protected areas worldwide (Joppa and Pfaff, 2009)). Additionally, remote areas tend

to lie in regions with less wealth, leading to an underrepresentation of research requiring cost-intensive equipment.

We therefore propose to adapt the above presented *in-situ* measurement systems to mixed systems, where sample equilibration occurs *in-situ* but analysis at a central laboratory.  This should be useful where *in-situ* measurements are impossible, due to lack of power supply and safe storage of equipment, or when large numbers of samples or simultaneous observation are a requirement.

Here, we introduce an adapted sampling method based on a vacuum pump powered by a 12V battery (derived from the borehole method by Marshall et al. (2020)) and a commercially available storage container (adapted from the SWIS System introduced by Havranek et al., (2020), making the presence of an IRIS in the field redundant. We tested our VSVS (Vapour Storage Vial System) using water sources of known isotopic composition in an extensive lab trial and added data from a field trial carried out in a boreal forest in northern Sweden, where we could test the suitability of the proposed method and

identify possible limitations. We include a section "preceding work" in the Results section to give the reader a chance to avoid repeating our failures if attempting to improve this methodology.

## 2 Material and Methods

### 2.1 VSVS lab test

We conducted a laboratory test with water of known isotopic composition (i.e. standards). The liquid standards (50 ml) were stored in 250 ml Duran® bottles (DWK Life Sciences, Staffordshire, UK) closed with a rubber stopper allowing repeated sampling. The sampling vials were 50 ml crimp neck vials (VWR1548-2092, VWR International AB, Stockholm, Sweden). The vials were dried in the oven at 65°C for 24h prior to use, and stored in a desiccator prior to crimping, to avoid atmospheric moisture to adhere to the walls as much as possible. They were then crimped using aluminium bands over lids

composed of a two-sided coating of polytetrafluoroethylene (PTFE) (inner) and butyl (outer) (SUPELCO SU860084, Merck, Darmstadt, Germany). Crimping (the tool works analogous to pliers) the vials was done carefully and each lid was doublechecked for position and tightness. Vials with a twistable lid were excluded from usage, to avoid atmospheric diffusion into or out of the vials.

  The lids ensured that the sample was in contact with only glass or PTFE (inner surface of lid). PTFE is a diffusion-tight

material, which is hydrophobic and chemically inert. It is recommended by the Picarro, Inc. (Santa Barbara, CA, USA) to use PTFE coated lids to store liquid samples, and is therefore, to date, the most suitable material to store water (vapor) samples if glass and stainless steel is unavailable.

  The outer seal made from butyl ensured air-tight re-sealing after sampling via a 0.7 mm needle. Subsequently, the vials were flushed with air containing equilibrated water vapour of known isotopic composition (hereafter referred to as "Source") for

10 min (see Fig. 1 for the setup) using the suction created by the cavity ring down spectrometer (CRDS, L2130-i, Picarro Inc., Santa Clara, CA, USA). Dry air was pulled from a laboratory gas drying unit (Drierite®, Fisher Scientific, UK), which dried the air down to 250 – 800 ppmV depending on room temperature. The dry air supply was connected using a silicone tube forced over PTFE tubing (1/4", Wolf Technik eK, Stuttgart, Germany) attached to a female luer-lock tube connector (CS - Chromatographie Service GmbH, Langerwehe, Germany) with an attached hollow needle (Henke-Ject®, 0.7x50mm,

Henke Sass Wolf, Tuttlingen, Germany) on the other end. The connection between the source and the sample vial was similar, but with needles attached to both ends, while the final connection between the sample vial and the CRDS consisted of a needle on one end of the tube and a stainless-steel fitting (1/4" Swagelok, Stockholm, Sweden) on the connection to the CRDS (Fig. 1).

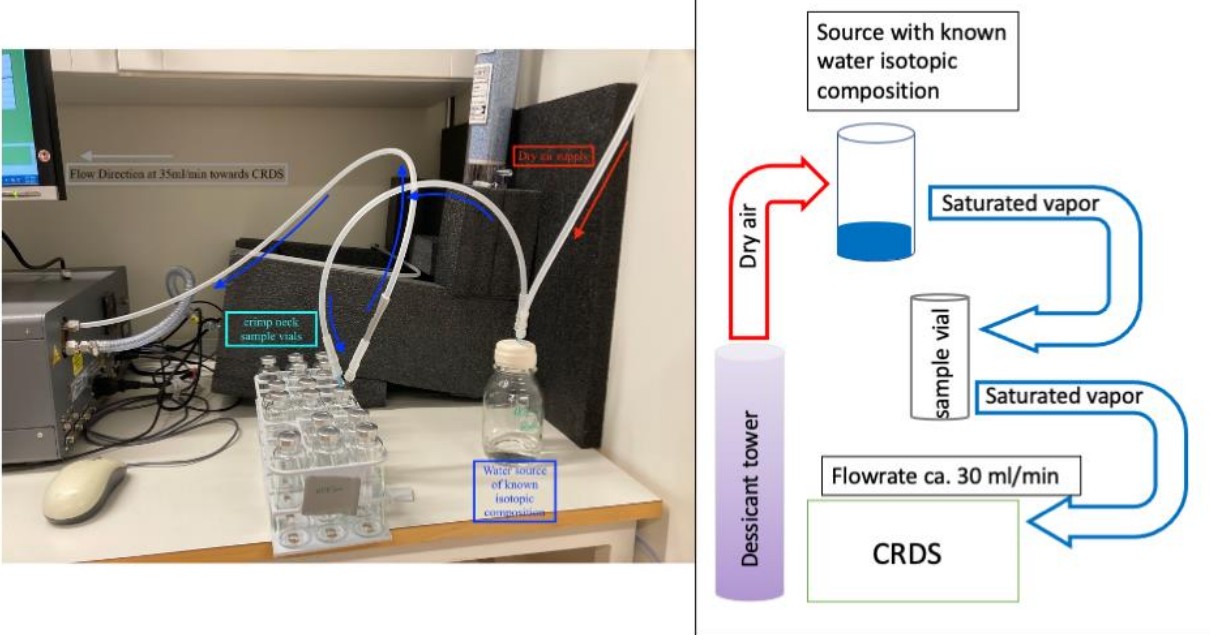

Fig. 1: Setup of sampling in the lab experiment. The CRDS creates suction from the headspace of a water source of known isotopic signature (dark blue) into the crimped vials (turquoise). Pressure deficit is compensated by air from a desiccant (Drierite®). The right side of the figure includes the same setup, this time as a schematic plan to improve readability. The isotopic composition and the water vapour concentration are monitored for 10 min before the vial is disconnected from the flow and stored for later analysis.

We monitored the water vapour concentration and isotopic composition as we flushed the sample to be able to detect the time when the water vapour concentration stabilised, which was after 8 min. After stabilisation, we flushed the samples for two more minutes to allow for one more complete exchange of the sample volume, leading to a total flushing time of 10 min and six complete turnovers. Since the flow rate created by the CRDS can vary between instruments (ours was ~30 ml/min), we advise the reader to carefully check the flow rate generated by their instrument and adapt the flushing time accordingly.

We selected five sources of water with different isotopic composition to test this method not only for natural abundance applications but also for examining the applicability for labelling studies, where water enriched in $^2$H is often used. Three of the sources covered large parts of the natural abundance range for precipitation composition (i.e. "light", "medium" and "heavy"), and two more artificially enriched sources covered much of the labelled range (i.e. "very heavy" and "crazy heavy" see Table 1). The isotopic composition of these sources was measured on the CRDS using an autosampler and calibrating the measurements against "in-house standards" ($\delta^2$H: -102.90, -64.01, -10.27, 53.89‰; $\delta^{18}$O: -25.13, -9.28, -5.22, -0.40‰). All isotopic compositions are reported in per mil (‰) relative to Vienna Standard Mean Ocean Water (VSMOW) (Eq.1):

$$\delta^2 H\text{‰} \; or \; \delta^{18}O\text{‰} = \left(\frac{R_{sample}}{R_{standard}} - 1\right) * 1000\text{‰} \,, \tag{1}$$

where R is the isotope ratio of the sample or the known reference (Craig, 1961b).

Replicated vials were stored for 0, 1, 3, 4, 7, and 14 days, where storage of 0 days means the samples were analysed the same day they were collected ("0-day" samples). Samples were kept in racks at room temperature in the lab. Each source and each storage time consisted of at least ten (five for the sources "heavy" and "very heavy") replicates. Before analysis, the racks with the samples were placed on a heating plate at 40°C for 10 min to reduce adsorption on the walls of the vials. The samples were then measured while standing on the heating plate.

For sample analysis, the dry air supply and the CRDS were directly connected to the vial. We let the CRDS pull the sample vapour from the vial at the same time as dry air replaced the now missing volume in the vial (at ~35 ml min$^{-1}$). This way the vapour concentration in the sample vial steadily decreased as the dry air diluted the water vapour. Because no water vapour was being added, the isotopic composition of the sample remained unaffected (see scheme in Fig. 2). Again, the vapour concentration and isotopic composition were monitored.

We excluded the initial isotope purge by calculating the slope of the vapour concentration over time. We filtered out all data before f'($dH_2O/dt$) = minimum slope, which marks the beginning of the recession curve unaffected by ambient air and thus corresponds to the plateaus in the isotope data (Fig. 2). We then calculated the mean isotopic composition from the two minutes starting from the time of the identified minimum slope (see yellow dots in Fig. 2B). We converted the vapour-phase measurements to liquid-phase data by assuming the vapour had been at equilibrium with the liquid water supply during

sampling using Majoubes' fractionation factors (Majoube, 1971) and source temperature measured with a commercially available thermometer (TFA Dostmann 30-1012).

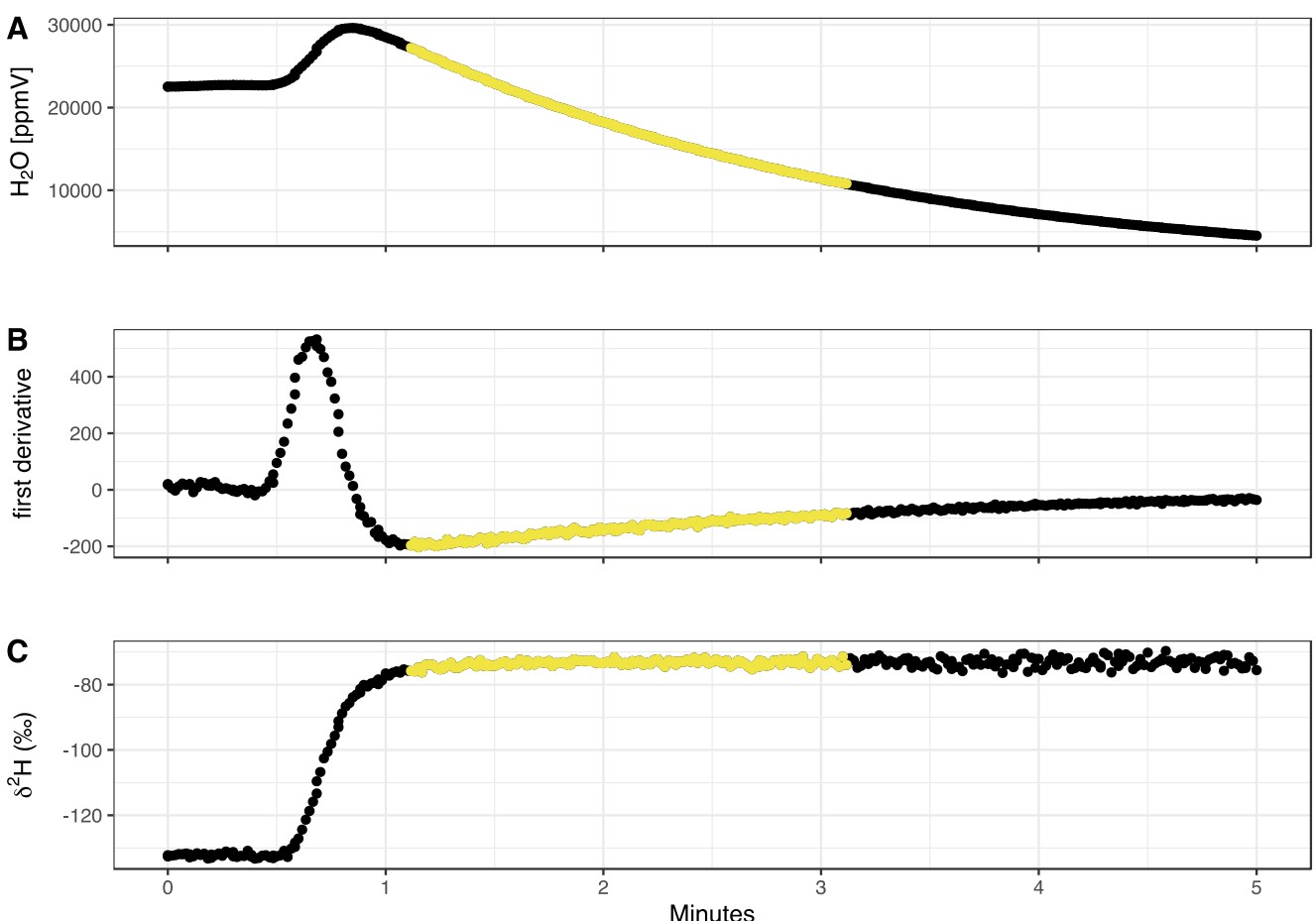

**Fig. 2 Exemplified measurement data for the "heavy" source over a time period of five minutes. Vapour concentration (A), first derivative of vapour concentration over time (B) and hydrogen isotopic composition (C). Following the criteria defined in the main text, we computed the mean of the isotopic composition ($\delta^2$H and $\delta^{18}$O) over the two minutes depicted by the yellow dots in the scheme.**

### 2.2 VSVS diffusive exchange during storage test

To show that the cleaning protocol and diffusive exchange into or out of the sample vials was negligible, we sampled dry air from the desiccant tower into three sampling vials and stored those for 14 days. We then measured the vials using the same setup as with the "real" samples. The analysis of these vials was done as described in the Supplement and the data are summarized in table S1.

If we assume that the tubes began the two-week test at ~600 ppmV (which is approx. what the dry air vapor concentration was on the sampling day) and ended it at ~1300 ppmV, then the leakage rate averaged 60 ppmV day$^{-1}$. This leakage rate would have negligible influence on the high water vapour concentrations (generally > 20000 ppmV) typical of our samples, but it reinforces the value of measuring the samples as soon as possible after they are collected.

For further testing whether diffusive exchange was affecting the isotopic composition of the stored vials, we measured the isotopic composition of the atmosphere during several days of the lab experiment. We expected diffusive exchange with the atmosphere to lead to altered isotopic compositions of the samples in the direction toward said atmospheric composition.

## 2.3    VSVS Field trial

We conducted our field test opportunistically during an ongoing tracer pulse-chase experiment. The pulse chase involved the addition of $^2$H-enriched water (~1800‰ $\delta^2$H) to an area of approx. 200 m$^2$ surrounding a set of mature trees in a spruce-pine forest in northern Sweden. Briefly, we monitored the isotopic composition of the xylem water of eight tree individuals (four *Picea abies* and four *Pinus sylvestris* trees) before and after application of the tracer for a total period of five weeks. We used the borehole equilibration approach as presented in Marshall et al. (2020). We drilled an 8 mm hole through each tree's stem, flushed it with acetone to reduce pitch production and, after four days, connected the outlet side of the borehole to a valve unit, a pump and finally a CRDS to monitor the H$_2$O concentration and isotopic composition. We refer to this setup with the term "*in-situ* system" from here on (Fig. 3). The data presented here were collected on the last day of said experiment on September 1$^{st}$, 2021, and five weeks after the initial installation of the borehole. We monitored a single Scots pine (*Pinus sylvestris*) using the VSVS. The same tree was connected to the *in-situ* system prior to our VSVS sampling. The selected tree was approx. 21.1 m high, had a diameter at breast height of 20.7 cm, and the borehole was installed ca. 40 cm aboveground, where the tree diameter was 21.6 cm. Because this method has now been tested several times (Marshall et al., 2020; Kühnhammer et al., 2021), we used the calibrated *in-situ* data as our "true isotopic composition" of the trees' xylem. The calibration for the *in-situ* data was conducted as described in Marshall et al. (2020). We then tested the new storage method against it. While switching the system from *in-situ* to VSVS, we checked whether we could visually detect resin/pitch in the borehole. As we did not observe any, we concluded that we could attach the tree to the VSVS without cleaning the borehole with acetone again.

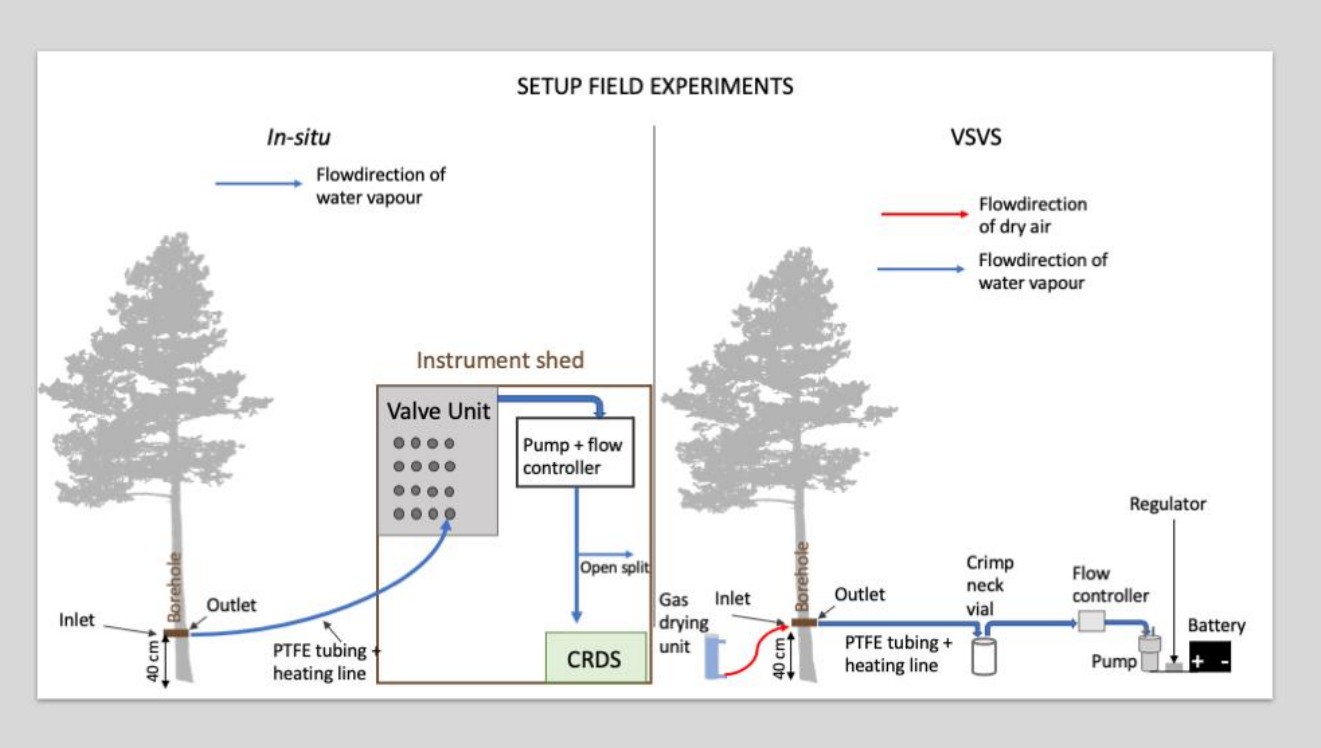

**Figure 3 Comparison of in-situ field setup (left) and sampling setup to obtain VSVS samples (right).**

Using the VSVS, samples were collected by connecting the "inlet" side of the borehole (in the original *in-situ* system this side was exposed to the atmosphere) to a gas-drying unit (Drierite®, Fisher Scientific, UK) and using a vacuum pump (no-name, 24 V, -50 kPa, https://www.ebay.de/itm/143587595483, last access 07/01/22) to draw saturated air from the "outlet" side of the borehole. In the original *in-situ* system this side was connected to the CRDS (see comparative scheme of the *in-situ* and VSVS setup in Fig. 3). The pump was connected to a power regulator and a mass flow controller (MFC, MC-2SLPM-D/5M, MCS-2SLPM-D-.25NPT/5M; Alicat Scientific, Inc., Tucson, AZ, USA), both powered by a rechargeable lithium-ion battery (12 V, 12 Ah). This battery is suitable for use in a remote area as it weighs less than 2 kg. With the present setup the pump and flow controller can run on the battery for more than 12 hours. The weight and dimensions of one carton (100 vials with lids) would be 30x13x50 cm and 2.5 kg, making transport to the field easy.

We set the flow rate of the MFC to 110 ml/min (which equals 77 $\mu$mol s$^{-1}$) to match the flow rate used in the *in-situ* system. According to the modelling exercise in Marshall et al., (2020) isotopic equilibrium is reached using flow rates up to 150 $\mu$mol s$^{-1}$ for trees of this diameter. As described in section 2.1 the vials were flushed for 10 min to allow the vial volume to be fully exchanged several times. The vials were filled sequentially such that all vials for "0-day" storage time were filled first, then all for one-day storage time, and so forth. While the vials were sampled, we continuously monitored the borehole temperature to be able to later convert the vapour-phase measurements to liquid-phase, again using Majoubes' fractionation factors (Majoube, 1971). To be able to do that, we had to ensure that no condensation would occur while the moist air was

pulled into the vial. This was accomplished by wrapping the PTFE tube with a heating line into foam insulation. The heating line was also powered by the battery.

Standards (i.e. the sources "light", "heavy" and "very heavy" from the lab test) were prepared in the same way as in the lab test, with the modification of the higher flow rate and using the pump in the field. All standards and samples were assigned to a storage group (i.e. 0, 1, 3, 7, 14 days). All samples were stored in the lab until analysis, except for the "0-day" samples, which were measured directly in the field three hours after sampling.

Measurements were conducted as previously described in section 2.1, with the modification of measuring each sample for only 3.5 min. This was done because on the day of the field trial the inside of the borehole was colder than the lab during the lab trial. The sampled air was therefore less moist, leading to lower water vapour mixing ratios (wvmr, in ppmV) in the vials. That meant the mixing with the dry air led to lower wvmr values more quickly than for the samples in the lab test, reducing the time period when wvmr were in the target range between ~17000 and 10000 ppmV $H_2O$. This concentration range was chosen to match the lab samples. We tried to avoid lower wvmr values as they generally associate with higher measurement uncertainties (https://www.picarro.com/products/l2130i_isotope_and_gas_concentration_analyzer, last access 07/01/22).

We switched from the battery-driven pump sampling to the *in-situ* system every four hours. Because the schedule of the *in-situ* setup measured this Pine tree every four hours, we were able to obtain 2 in-situ measurements during the VSVS sampling day (one at 10am and again at 2pm → n=2). As noted above, we compared the VSVS samples to the calibrated *in-situ* system data, which were considered our "gold standard". We disconnected the tree from the *in-situ* measurement system when not measured and re-connected it to the *in-situ* system 20 minutes before its measurement was scheduled. In between we sampled the equilibrated vapour as described above (five per storage group). We compared the "0-day" samples (n = 5) to the *in-situ* measurements of the same day (n = 2).

## 2.4    Analysis and Statistics

Calculations as well as graphical representations were conducted using the "tidyverse" packages in R (Wickham et al., 2019; R Core Team, 2020). To assess the VSVS's suitability to reliably store collected water vapour (assessing the "storage effect"), we calculated the change in isotopic composition (see Eq. 2 for either $\Delta\delta^2H$ or $\Delta\delta^{18}O$) over the storage time (t), relative to the mean of the "0-day" sample (t=0) for each source and for the lab and field test, respectively (Eq.2):

$$\Delta\delta = \delta_t - \delta_{t=0}, \tag{2}$$

This "storage effect" was then related to the storage period using a linear regression model, separately for oxygen and hydrogen as well as for natural abundance and enriched sources. The data were then corrected according to the storage period. Here, we provided an extensive data set (i.e. 5 different water sources with 10 replicates each, providing 50 datapoints per storage time), however, we encourage each group to create their own storage correction coefficients to be able to individualise it for their CRDS instrument.

We used the same model coefficients determined from the lab data to correct the field data samples. We additionally calculated the mean for each storage group (by source) and conducted pairwise Wilcoxon tests between the "0-day" samples and every other storage group, to disentangle effects introduced by the sampling method from storage. A Wilcoxon test is a non-parametric approach to detect differences between two groups of data, which are not normally distributed. Wilcoxon tests were conducted using the "*compare_means*" function of the "ggpubr" package in R (Kassambara, 2020).

To relate measurements to the liquid true values we used a linear regression model for each storage group using the "lme4" package (Bates et al., 2015). We used three-point calibration for both $\delta^2H$ and $\delta^{18}O$. That meant we separated the highly enriched sources from the natural abundance for $\delta^2H$, using the "heavy" source as the lowest standard for the enriched scale and as the highest for the natural abundance scale. The idea was to avoid "overweighting" the lower end of the enriched scale by adding three natural abundance standards to it.

## 2.5    Preceding work

The first tests for this method originate from a field trial in a boreal forest, where some of the authors attempted to trace an enriched water pulse through 120 trees simultaneously. Briefly, a hole was drilled through the entire diameter of a tree stem, equipped with brass fittings (Ahlsell AB, Sweden), and sealed from the atmosphere using chlorol-butyl septa (Exetainer, UK). Syringes (Henke Sass Wolf, Tuttlingen, Germany) were used to draw out 20 ml of equilibrated xylem sap vapour and the isotopic composition was subsequently measured on a CRDS via injection into a dry air stream (Magh et al., 2021). The time between sampling and measurement varied between 20 minutes and up to 5 hours.

We noticed that the water concentration and isotopic composition of the vapour in the syringes were altered within hours after sampling. Though the test revealed suitability for heavy label detection studies where e.g., response times revealed by isotope dynamics rather than absolute values may be of prime interest. However, we do not recommend using plastic syringes for long-term storage or for natural-abundance studies.

When developing the presented method further, we also tested crimp neck vials of 20 ml volumes, which would be even easier to transport and handle. However, after the first rounds of testing, we discovered that the volume was not large enough to give a stable 2-minute isotope plateau when measuring, so we discarded the idea of using vials smaller than 50 ml.

## 3 Results

### 3.1 Lab Test

Table 1 shows the mean and the variation occurring immediately after the vials were filled ("0-day" samples). These data give an overview of the minimum possible variation (method precision) during the sampling procedure and compare it to the expected values defined through the measurement of the liquid source on the CRDS. Results depended on the source sampled (see sd values in Table 1), indicating that the vapour sampling procedure introduces higher variation than the liquid phase measurements (Table 1).

**Table 1 Means and standard deviation (sd) for the five sources observed immediately after the vials were filled ("0-day" samples), as well as values for $\delta^2H$ and $\delta^{18}O$ determined by liquid water measurements on a CRDS. Mean for "0-day" samples derived from 10 replicates (n=10), values for direct analysis derived from 10 injections in the "high-precision" mode of the CRDS.**

| ID | $\delta^2H$ mean 0-day sample VSVS (‰ ± sd) | $\delta^{18}O$ mean 0-day sample VSVS (‰ ± sd) | $\delta^2H$ direct analysis (‰ ± sd) | $\delta^{18}O$ direct analysis (‰ ± sd) |
|---|---|---|---|---|
| **light** | -87.49 (3.9) | -12.03 (0.6) | -92.88 (0.05) | -12.74 (0.04) |
| **medium** | -53.80 (4.4) | -8.02 (0.8) | -52.25 (0.1) | -7.8 (0.03) |
| **heavy** | 1.10 (0.6) | -5.37 (0.1) | 0.88 (0.05) | -5.71 (0.005) |
| **very heavy** | 729.88 (4.3) | -11.60 (0.4) | 758.71 (0.3) | -12.34 (0.04) |
| **crazy heavy** | 1590.49 (65.8) | -10.31 (0.3) | 1728.31 (1.4) | -10.95 (0.03) |

We observed two different patterns between $\delta^2H$ and $\delta^{18}O$. We first address the storage effect on $\delta^2H$ and then on $\delta^{18}O$. Both patterns can be observed in Fig. 4, which compares the changes in $\delta^2H$ and $\delta^{18}O$ of samples stored for several days ($\Delta\delta$). Each storage group is compared to the "0-day" sample group using a Wilcoxon-test (see supplemental Table S2). The change in isotopic composition depends not only on the storage time but also on the enrichment in $\delta^2H$. The data show no consistent pattern regarding $\delta^2H$ over storage times on the natural abundance range (Fig. 4 A). The median change ranges from 0 to 5‰ (Fig. 4A, Table S2). This observation is further supported by the linear regression model relating the change in $\delta^2H$ to the storage period, as the slope of the fit is 0.06 and this model is not statistically significant (Fig. 5C), indicating that there was no storage effect. However, for the sources enriched in $\delta^2H$ the pattern reveals a constant depletion in $\delta^2H$ over time. The median change for the enriched sources ranges from -10 to +17‰ for $\delta^2H$ on storage day 1 and increases after that (Fig. 4B, Table S2). This is also described by the linear regression model fitting the change in isotopic composition over the storage period ($R^2$=0.11, slope -3.67‰ day$^{-1}$, p≤0.05, Fig. 5C).

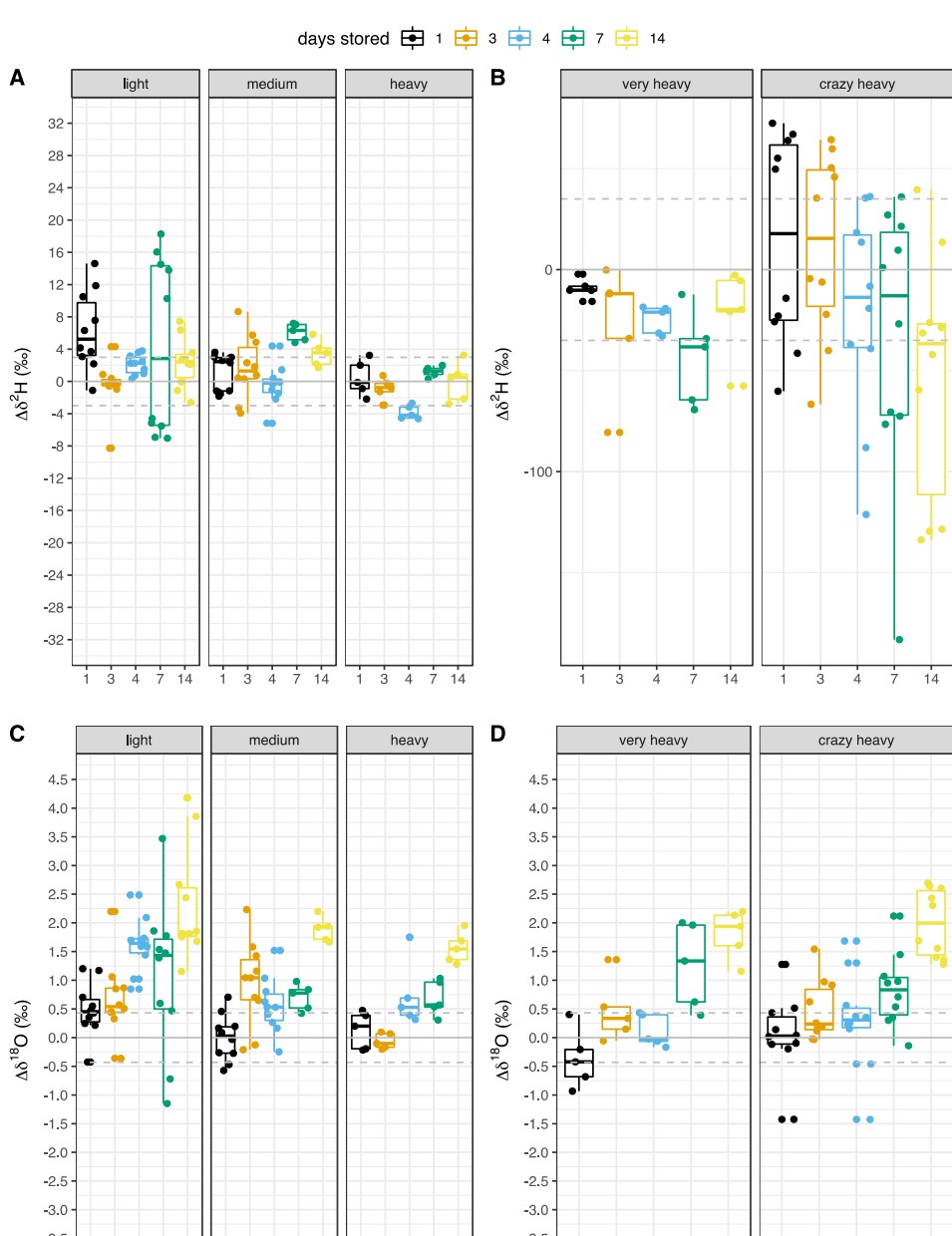

**Fig. 4 (A, B) δ²H and (C, D) δ¹⁸O change from the "0-day" samples over storage times. Each measured source is represented by a facet and storage time is indicated by colour (see legend). Natural abundance sources are depicted in panels A and C, while the artificially enriched sources are represented in panels B and D. Data are represented by boxplots with the box showing the median as a line, the data between the 25th and 75th percentile as the box, the whiskers representing the minimum and maximum of the data. The dots indicate each data point measured and the horizontal lines represent the mean (solid) and standard deviation (dashed) of the "0-day" samples. Note the scale change in panel B.**

285

Looking at $\delta^{18}O$, the enrichment consistently increased with increasing storage period (Fig. 4 C and D, Table S2). The "storage effect" was well described by a linear model using the change in $\delta^{18}O$ of all sources over the storage period. It is statistically significant and yields an $R^2$ of 0.43 (Fig. 5A). We additionally compared these "full" models to linear regression models for each source separately but did not observe any significant differences in the model slopes (Table S4). We therefore decided to use the "full" models (Fig. 5) to correct the storage effect.

The global meteoric water line reveals a tight relation between $\delta^2H$ and $\delta^{18}O$ with a linear fit and a slope of 8 (Craig, 1961a). Thus, scales for the natural abundance sources in Fig. 4 were chosen to be eight times greater for $\delta^2H$ than for $\delta^{18}O$ to enable direct visual comparison of the storage influence on the composition. This was slightly bigger for $\delta^{18}O$ as can be seen from Fig. 4, indicating less influence of storage on $\delta^2H$, which in turn, is also supported by the poor model fit (Fig. 5).

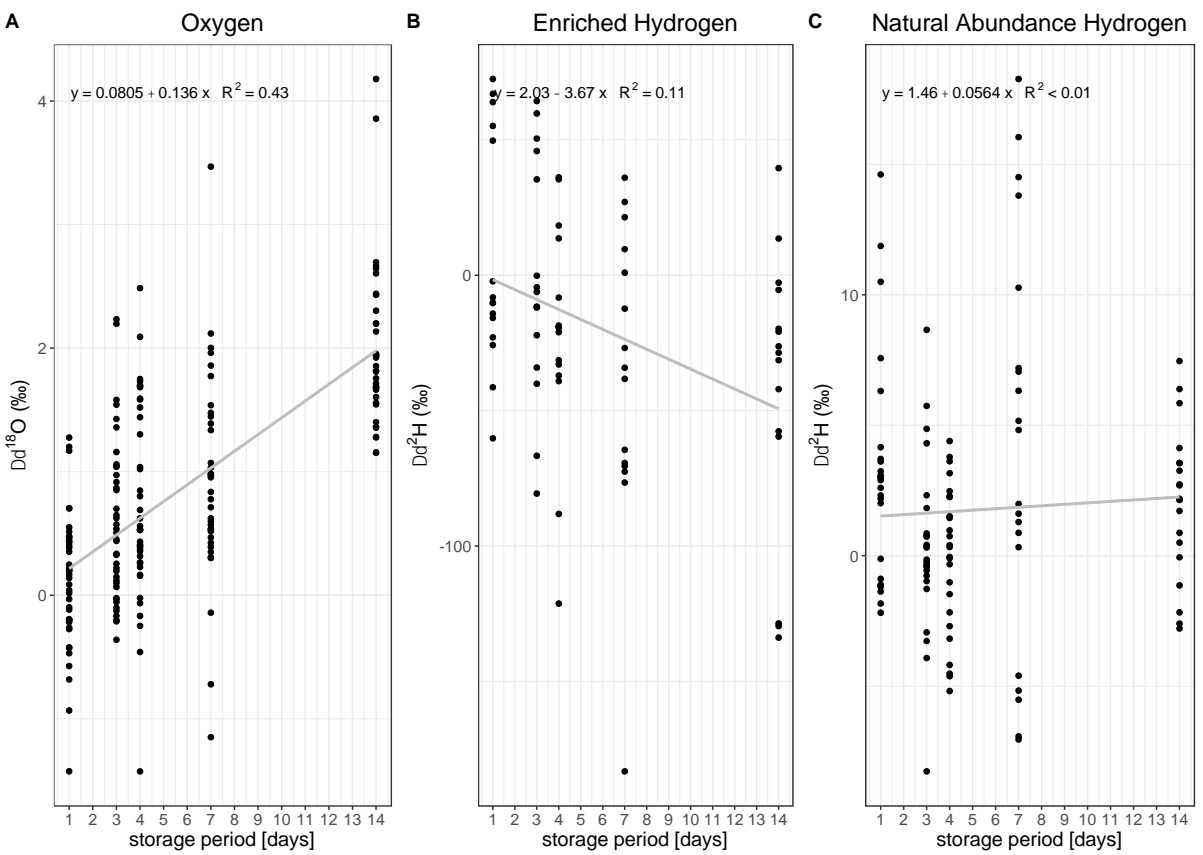

**Fig. 5 Linear regressions for the change ($\Delta$) in isotopic composition over the storage time. The analysis was conducted separately for $\delta^{18}O$ (A) and $\delta^2H$ (B, for enriched in $^2H$) and $\delta^2H$ (C, for natural abundance). The change in isotopic composition was significant for $^{18}O$ (see panel A for regression equations and $R^2$), and $^2H$ (panel B) for the enriched sources. It was insignificant for $^2H$ change in the natural abundance sources (panel C).**

We then analysed the uncertainty of the stored vapour samples based on their true liquid isotopic composition. We used linear regression models for three natural abundance sources ("light", "medium", "heavy") and for three for the enriched

sources ("heavy", "very heavy", "crazy heavy") for δ²H, and all natural abundance sources for δ¹⁸O, at each storage time (Fig. 6). Overall, the model fits for δ²H are better than for δ¹⁸O, though both show high $R^2_{adj}$ values. The high $R^2_{adj}$ indicates that they are sufficient for empirical correction. The linear relationship between the liquid water and the measured vapour isotopic composition was statistically significant for all storage times ($p < 0.01$ for 10 samples per source and storage day), with similar slopes (Fig. 6). Though the slopes were similar we intended for the option to calibrate each set of samples with

their respective slope and intercept.

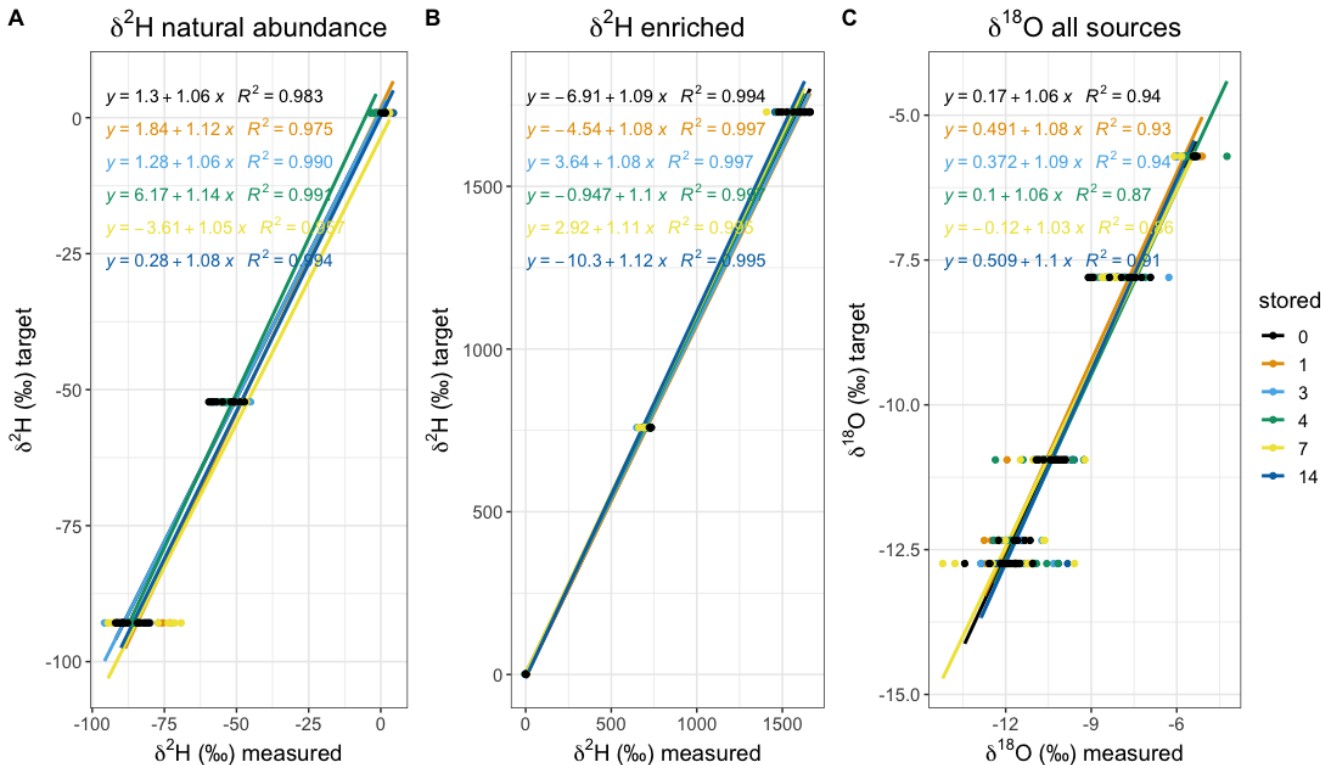

**Fig. 6 Linear regression through δ²H of the natural abundance (A), enriched (B) and δ¹⁸O (C) sources grouped by storage time (days) for the lab trial. All models are significant on a level α=0.05. The regression equation and R₂ are plotted in the colour of the datapoints and the fit (see legend for color-coding).**

The calibrated and uncalibrated data can be derived from Table S3 and are plotted in Fig. 7, showing that storage-effect correction and calibration reduces the variability between the storage groups, moving the samples close to their true liquid value.

We recorded the atmosphere's isotopic composition during sampling and the measurement days to check for admixture of

the atmosphere into the vial during storage (dashed blue lines in Fig. 7). We were thus able to rule out diffusion of atmosphere into the vials as all three standard sources would have been altered towards the atmospheric composition. This

would have led to depletion rather than enrichment of heavy isotopes with increased storage periods, which was not the case (Fig. 4, 7 and Table S2).

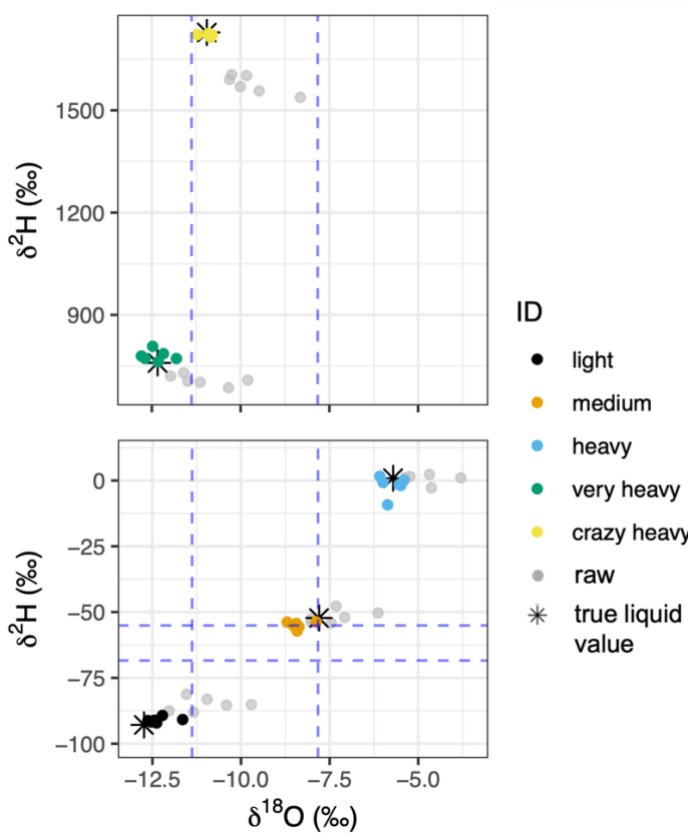

**Fig. 7 Dual isotope plots of the raw mean (grey dots) and corrected/calibrated mean (coloured dots) of the lab trial storage data, corrected for the" storage effect" and calibrated using the linear regression models of each storage time. Sources are depicted by colour and the liquid true value is indicated by black stars. The upper panel shows the data of the two sources enriched in $^2$H ("very heavy" and "crazy heavy", for the calibration we also used the "heavy" source, however we refrain from plotting it again here as it unnecessarily enlarges the Figure), while the lower panel depicts the three natural abundance sources (i.e. "light", "medium" and "heavy"). The dashed blue lines show the range of the background atmosphere during a few days of the lab trial. They are indicated here in support of the argument of isotopic non-drifting towards the room air (see text).**

### 3.2 Field trial results

The values of the VSVS samples were generally similar to the mean of the *in-situ* samples. The *in-situ* data revealed stable $\delta^{18}$O values (-13.15 ± 0.01‰) throughout the day, while *in-situ* $\delta^2$H varied up to 3.2‰ from a mean of 1.7‰ (Fig. 8). During the in-situ measurements we continuously monitored the "CH4" variable recorded simultaneously on the CRDS to check for potential spectral contamination associated with organics originating within the borehole. The data did not reveal any differences in that variable between measuring standards (no organics) and the boreholes (potentially organics) (data not shown). There were significant differences after some storage times as was indicated by Wilcox test (Fig.8). The VSVS data

failed to return the *in-situ* $\delta^{18}O$ when comparing the corrected and calibrated $\delta^{18}O$ of the "0-day" sample to the *in-situ* measurements (Fig. 8, Table S5). They became enriched relative to the source over longer storage times. In contrast, the VSVS $\delta^2H$ data, as already observed in the lab trial, did not follow a constant pattern. VSVS samples stored for one and seven days did not differ significantly from the *in-situ* measurements (Fig. 8).

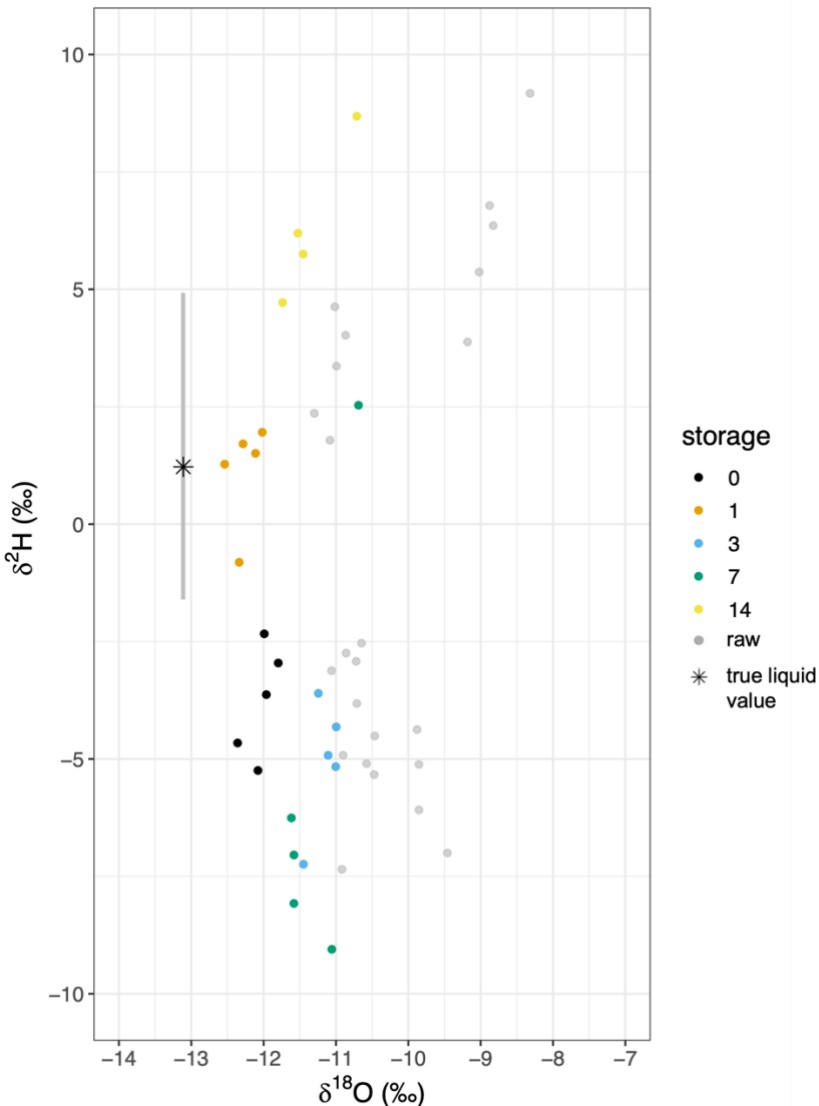

**Fig. 8 Comparison of *in-situ* and corrected VSVS data by storage time for (A) $\delta^2H$ and (B) $\delta^{18}O$. *In-situ* data are depicted in black, while VSVS samples are indicated by colour. Wilcox test identified significant differences between the *in-situ* data and each storage group (differences are indicated by asterisks: "*" indicates p<=0.05 and "ns" not significant).**

## 4       Discussion

We performed a lab trial of a water-vapour storage method using water sources covering stable isotope ratios in the natural abundance range and well beyond it into the range highly enriched in $^2$H. This was done to test the suitability of an *in-situ*
approach to capture and reliably store water vapour combined with lab methods to analyse it. We added data from a field trial to further test the method's applicability under field conditions. We then compared the VSVS field data to the *in-situ* field data, the latter providing our "gold-standard". Overall, we found the method to perform well in the lab while in the field it performed within a defined range of precision and storage time.

### 4.1       Suitability of sampling method

We show that our adaptation of the *in-situ* method (Marshall et al., 2020) can simplify the analysis while reliably reproducing the isotopic composition of natural abundance samples when measured on the same day. The method is robust and cost efficient as it uses only a battery-powered pump and a flow controller to collect the samples, then the water vapour is stored in commercially available crimp vials, which can be re-used.

The reproducibility of measurements lies within the range reported for other *in-situ* approaches, e.g., Volkmann et al., (2016a). For example, the median reproducibility was 2.8‰ for $\delta^2$H and 0.33‰ for $\delta^{18}$O, while the uncertainty was up to 20‰ for $\delta^2$H and 3‰ for $\delta^{18}$O (Volkmann et al., 2016a; Beyer et al., 2020). In Marshall et al., (2020) the authors found their measurement precision to range from 2.3‰ to 7.8‰ for $\delta^2$H from natural abundance towards mild enrichment. For $\delta^{18}$O it ranged from 0.22‰ to 0.6‰. Thus, the VSVS provides a possible solution for settings where tree numbers are large, sampling sites lie far apart, or laboratory facilities are at some distance.

Although we were unable to reproduce the standard value within the above range for the samples highly enriched in $^2$H (over 1500‰, "crazy heavy"), we do not regard this as surprising. High enrichment is generally associated with lower precision and samples outside the VSMOW-SLAP range cannot be calibrated to the same uncertainty level as samples within that range since in this case the requirement of "bracketing" samples with standards cannot be met. In labelling studies, the signal is usually so strong that higher noise can be tolerated.

However, we point out that our sampling method did not reliably reproduce $\delta^{18}$O of the vials sampled in the field trial as it was about 1‰ more enriched than what the *in-situ* measurements suggested and showed considerable variation in $\delta^2$H. We treat this result carefully as this was only the case for the samples but not the standards sampled with the same method in the field and at the same time call for further studies investigating the VSVS field suitability and in this context adding information about different water conditions in the tree (i.e. stressed vs. non-stressed) and see if and how that is influenced by storage.

## 4.2 Storage period significantly influences isotopic composition

In the crimp neck vials we observed a significant change in isotopic composition over time. The direction of change for $\delta^{18}O$ was constant enrichment with longer storage time, indicating possible exchange with the atmosphere. The most reasonable explanation in this context is leakage through the lid from the higher water concentrations inside the vial towards the lower concentrations on the outside. This is the best supported scenario, as the data supports that there is negligible leakage.

Changes in $\delta^2H$ were inconsistent and less pronounced than for $\delta^{18}O$. In most cases $\delta^2H$ became significantly different after three days of storage, meaning that after three days the analytical range of variation was exceeded. For an overnight storage experiment with the "SWISS" system, Havranek et al., (2020) reported changes in isotopic composition between 0 and 1‰ for $\delta^{18}O$ and between 0.3 and 4‰ for $\delta^2H$. Our data suggest changes between 0 and 0.5‰ for $\delta^{18}O$ and between 0.4 and 6‰ for $\delta^2H$ when considering the natural abundance range. This indicates that for overnight storage both the VSVS and the SWISS perform on a similar level. However, when comparing our longest storage period (i.e. 14 days) to the 24-day storage in Havranek et al. (2020), it becomes clear that the VSVS does not sufficiently preserve the isotopic composition of its contents during the experiment, while the SWISS continued to perform accurately. For $\delta^2H$ however, we found mean changes between 0 and 3.4‰ after 14 days for the natural abundance samples, which can be considered sufficiently small depending on the research question. Nevertheless, we do recommend measuring samples within three days after sampling to get a result within the error margin of the methodologically introduced variation.

In addition to the smaller sensitivity of $\delta^2H$ towards storage in the VSVS, we further recommend it for its low cost (Table 2). As noted above, all components are available off-the-shelf and, as of this writing, one litre of 99.9% $^2H_2^{16}O$ costs ~1000€, while one gram of $^1H_2^{18}O$ costs ~330€.

## 4.3 Isotopic changes with storage period can be corrected using linear models

Given the potentially significant, yet systematic shift in VSVS data over time, we strongly recommend to prepare standards within the "equal treatment" framework as emphasised in e.g., Gralher et al., (2021). This means that the standards are sampled on the same day as the samples, stored under the same conditions and for the same period of time. One can then presume that any systematic, storage time related isotopic shift in the samples is matched by the standards. Using this approach, we gained higher precision and accuracy for both lab- and field-based data.

For the field data set we emphasise the potential for additional variation due to the trees' water use and transport. As the sampled tree was constantly transpiring water throughout the sampling process and the sampling took roughly 50 min per storage group (for each sample, 10 min x 5 replications = 50 min), variations in the data may have originated from true variation of xylem isotopic composition. Differences in the trees' xylem water isotopic composition over the course of day have previously been observed and described in, e.g., De Deurwaerder et al., (2020). In future studies this could potentially

be avoided by reducing the sampling time per sample. The 10-minute sampling interval used here was derived from the low
flow conditions of the CRDS in the lab trial, while the higher flow rate in the field trial would allow for shorter sampling
times at the same sampling precision.

## 4.4    Time and Cost Efforts

To be able to make an informed decision about costs and time effort regarding the VSVS, we compare the VSVS to an *in-situ* system and destructive sampling and subsequent extraction via a cryogenic extraction line after (Koeniger et al., 2011)
(Table 2). The data for the latter two have been obtained from Kübert et al. (2020). Each method has its own advantages and disadvantages. In terms of equipment costs, the VSVS is the cheapest, even when including the running costs for repeatedly buying new lids and needles. In terms of time effort, the *in-situ* system and the VSVS are more efficient than obtaining and analysing samples for the cryogenic extraction line. Overall, the VSVS combines cost and time efficiency when compared to the two alternatives.

**Table 2 Cost [€] and time [h] effort overview comparing *in-situ* systems, cryogenic extraction and VSVS. * per 100 samples ** per installed soil depth in Kübert et al. (2020), *** including 6 injections per sample on a CRDS. Data for the *In-situ* and Cryogenic Extraction analysis derived from Kübert et al. (2020). The time effort includes the time needed for setup, and maintenance but not for data analysis. The costs for power and gas supply are not included.**

|  | *In-situ* | Cryo. Extraction | VSVS |
|---|---|---|---|
| Equipment | 1775 | 8000 | 625 |
| Tubing | 14.2[**] | - | 15 |
| Time [h][*] | 25 | 60[***] | 25 |
| Running Costs[*] | Almost none | 120 | 50 |
| Know-how for setup and handling | Medium-difficult | Medium | Low-medium |

## 5    Conclusions

We introduced and tested a simple and cost-efficient approach to sample and store water vapour to enable plant or soil water isotope measurements that does not require access to line power. We proved the suitability of the sampling method within an extended precision range for natural abundance and samples heavily enriched in $^2$H. We successfully tested the approach both in the lab and in the field. The isotopic composition of water was not significantly altered in a storage time of 3 days for $\delta^{18}$O and $\delta^2$H was not altered beyond the variation introduced initially by sampling. This method extends the utility of *in-situ* sampling of water vapour, simplifying the collection and measurement of samples from which the isotopic composition of liquid water sources can be inferred.

## Code/Data Availability

Code and Data are available from the corresponding author upon request.

## Author contributions

RKM and JM designed the experiments. RKM and AK conducted the experiments. RKM and HL performed the data analysis. RKM, BG, BH and JM conceived the theoretical parts. RKM wrote the first draft of the manuscript and implemented the revisions together with JM. All authors contributed with advice, prepared and reviewed the manuscript.

## Acknowledgments

We would like to thank Jose Gutierrez Lopez and Hjalmar Laudon for support during the field campaign. Special thanks also to Jonas Lundholm for providing advice and support in the lab. RKM and JM were supported through grants from the Knut and Alice Wallenberg foundation: KAW 2018.0259 (RKM and JM) and KAW 2015.0047 (JM). AK was funded by a "Short Term Scientific Mission (STSM)" grant provided by the COST Action (CA19120) WATer isotopeS in the critical zONe (WATSON).

## Competing interests

The authors declare that they have no conflict of interest.

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
