# Peer review of "Technical note: Conservative storage of water vapour: practical *in-situ* sampling of stable isotopes in tree stems"

_Hydrology and Earth System Sciences, 2022_

## Author Comment (AC1)

Comments/Answers community comment:

Magh et al. are investigating, if equilibrated water vapor from soils and plants can be collected and be measured thereafter in the laboratory in order to determine water isotope values. The results of their experiments demonstrate that this is possible within an acceptable uncertainty compared to in-situ approaches  (ll. 22-23, please rephrase this sentence so that it is clear to what this uncertainty refers).

We will rephrase the sentence in the revised manuscript to ensure readers understand it refers to a higher acceptable standard deviation from the mean associated with the higher measurement value of the enriched samples.

Having applied and developed in-situ methods since 2016, I applaud the authors for proposing a method to overcome one key limitation related to in-situ approaches: The spatial resolution. Having a laser spectrometer in the field is expensive, risky; and direct measurements is extremely laborious and error-prone. Hence, this can be a first step towards enabling the full range of benefits of in-situ measurements: high spatiotemporal resolution and measurement of plant-available or mobile water.

While the method is carefully tested in this manuscript, a number of aspects remain to be tested, e.g. application in real field environments, temperature fluctuations (e.g. sample transport in an airplane), maximum storage time, test of different flow rates for equilibrating the sample in the field, compare Marshall et al., 2020; , carrier gas to be used (maybe using a dessiccant tower would be sufficient in the field, where dry air is not always available?). The remaining shortcomings and potential factors that could affect the method could be pointed out more clearly at the end of the manuscript.

We agree to add a section to the revised manuscript where we will pinpoint these shortcomings of the current manuscript and emphasize the potential for further research using this method.

An option that is not discussed is having the instrument in the field (but in a 'safe' space) or nearby, and measure the samples directly in the field, but not via connectors etc. This would limit sample storage time and perhaps guarantee best results. For instance, we are testing the water vapor storage method at a site in central America in a setting where the next isotope laboratory is 4 driving hours away; this is a potential setting that many might have. How will altitude/pressure differences and temperature alterations affect the storage? The risk of this method is clearly the small amount of water molecules stored in the bottles, which makes it very easy to be contaminated.

We agree, the possibility of having a Laser close-by is an option that is not extensively discussed, and it will be added to the discussion section in the revised

manuscript. We will also address the risks involved in transport more clearly but think we have addressed a potential solution for at least condensation (which would be the worst case in most settings) by heating the samples prior and during the measurement procedure. We will make sure this is clearly emphasized in the revised version.

In my opinion, the title could be more concise and related clearly to in-situ measurements of water isotopes (e.g. by mentioning in-situ in the title, it will increase the visibility of the manuscript imo).

We disagree with you here, since this method would technically not only enable in-situ measurements in the sense the term is used in the ecohydro community but also enable sampling of air if moisture is high enough. On the other hand, you're right, adding the term in-situ would probably enhance the visibility of the manuscript. We will reconsider the title in the revised manuscript.

I strongly recommend this experiments to be published in HESS and thank the authors for sharing this work.

Kind regards,

Matthias Beyer

---

## Author Comment (AC2)

**Comment and Answers on hess-2022-37**

Rachel Havranek (Referee)

Referee comment on "Technical note: Conservative storage of water vapour: a key to practical measurements of water stable isotopes in tree stems and soils" by Ruth Magh et al., Hydrol. Earth Syst. Sci. Discuss., https://doi.org/10.5194/hess-2022-37-RC1, 2022

**General comments:**

In this paper, the authors test a flexible, cost effective way to sample water vapor from trees for stable isotope geochemistry. This kind of system fills a strong need for the stable isotope community, and will be very useful for many different applications. To test that their system was reliable, the authors performed storage tests both in the lab and in the field. The authors found that there was systemic storage bias in oxygen isotopes over time, and that bias was only present for 'crazy heavy' waters for hydrogen isotopes over time. Below, I suggest the addition of one simple experiment to the manuscript to demonstrate that the vial cleaning protocol is sufficient, and that the vials are sufficiently resistant to atmospheric intrusion over the proposed storage timescale (3 days or less). Broadly, I think this is an excellent paper and the system will be widely used by the community. I strongly support the publication of this paper in HESS.

Rachel Havranek

We thank Rachel Havranek for this very comprehensive and insightful review and hope to answer all her comments/questions sufficiently and revise the manuscript accordingly.

As for her main concern about the cleaning and storage sufficiency, we agree to add data to the revised manuscript, which we already obtained for the original draft. We will include those in the revised version.

**Specific Comments:**

***Atmospheric intrusion & vial cleaning protocol***: My largest comment on this paper is that the authors did not sufficiently address the issue of atmospheric intrusion, nor did they demonstrate that they sufficiently eliminated an atmospheric signal from their vials prior to sampling.

We were recently made aware that atmospheric intrusion in the soil community is referring to the mixing of atmospheric air into the subsurface (see e.g. (Lowrey et al. 2016)). To avoid confusion between the communities and because this method can be used for soil vapor sampling, we will rephrase the respective section in the revised manuscript and will from now on use the term "diffusive exchange" when talking about the exchange between vapor inside the vial with the atmosphere.

As mentioned above, we do have the data showing that cleaning protocol and diffusive exchange in and out of the vial are highly unlikely and will include those in the revised version of the manuscript.

Briefly, we sampled dry air from the desiccation tower into three vials and stored them for 14 days. We then measured the vials the same way as the other samples and analyzed the data accordingly. We did measure water vapor contents ranging in the same magnitude as when

measuring the desiccation tower directly. However, we need to emphasize here (and will do so in the revised manuscript) that the "dryness" of the desiccation tower varies with temperature (which is to be expected) and can thus only give a range between 300 and 600 ppmV for that value. The samples we measured lay within this range, which lets us conclude that the signature within the vial prior to sampling is sufficiently "overwritten" (making the cleaning protocol sufficient) and supports our previous argument that we cannot observe atmospheric intrusion into the vial even during 14 days of storage.

With regards to the cleaning protocol: I appreciated the discussion of vial cleaning protocol and I think that baking the glass vials at 65°C for 24H is likely sufficient. However, I have concerns that the PTFE caps were sufficiently dried (since PTFE is SO 'sticky'), and suspect that might the source of *some* of the observed drift. When I have played with PTFE fittings in the SWISS system, I've been disappointed by how much PTFE exchanges. I am also curious what gas the vials were purged with prior to sampling? If there is atmosphere in the vials when they are crimped, and then they cool post heating, I would expect atmosphere to stick to walls of the vial, which could ultimately exchange with sample vapor, even after so many vapor 'turns' during sampling.

We are a bit surprised about the "stickiness" comment regarding PTFE. This is not what we, nor other authors working in the same field observed when using PTFE tubing or lids. Picarro recommends Teflon coated lids to store liquid samples because of its properties (see here: https://www.enviscid.com/uploads/5/6/3/8/56382687/l2130-i_users_manual_rev_a.pdf). We will extend the discussion of the revised manuscript a little, to show that PTFE is used because it is the most diffusion tight material available on the market, it is chemically inert and best option when stainless steel/glass is unavailable or impractical. Properties say it's hydrophobic therefore stickiness would not be easily explicable.

As mentioned above, we can also show with the newly available data that the sample turnover does sufficiently overwrite whatever leftover atmosphere is in the vial. Also, the vials are not purged with any gas prior to sampling and kept in a desiccator until cooled off. This information will be added to the protocol in the revised version as it is currently missing.

I also think that the authors need to do a little more work to demonstrate to the readers that atmospheric intrusion is not a source of error for this system. The authors didn't include data in this initial submission to allow readers to evaluate if that was a source of error. This would be a simple and convincing addition.

We agree and will therefore add the samples that contained dry air only, and additionally data on the atmospheric isotope composition, towards which the samples should have drifted had there been diffusion into the vial from the atmosphere.

Encouragingly, for most natural waters the VSVS system was within error of direct measurement. What worries me is that 0-day did not overlap within uncertainty for either oxygen or hydrogen for the light standard – which is a pretty typical high altitude and/or high latitude value. It's not clear to me that method precision is truly accounted for. To hit two birds with one stone, I suggest a very simple, short timescale experiment where the authors fill a set of vials (perhaps 10 to sufficiently catch crimping variability?) with just dry air from the drierite system, and then do a storage test, just as they did with the rest of their lab tests. Given the authors' recommendations in the discussion, I think a 3 – 5 day experiment should be sufficient. I would also recommend that the authors measure the dry air the day they fill the vials so that small changes in water

concentration can be detected. With our drierite system, I know that there can be some variability and so it's nice to have that baseline value recorded.

See above answers.

***Other protocol questions***: Given that this paper is directed towards an audience of potential future users, I have a few small questions about lab protocols that could likely be answered either through some supplemental text or the addition of a few short sentences into the main body of the manuscript

Crimping: I am unfamiliar with how the crimping process worked, I think a very short (a few sentences at most) discussion of how to know that a cap has been sufficiently crimped or has been over-crimped (and therefore leaky) would be very helpful for the target audience. Alternatively, is there a way the authors imagine they could screen for that during sample measurement?

Very good idea, thanks. Yes, the crimping process is a potential error source e.g. when the crimping tool (which works similarly to pliers) pinches too hard or not enough. The handling person is then able to turn the lid around the bottle neck, which should be avoided. We did not have a problem with this but acknowledge this could be a problem.

We will add this to the revised manuscript.

Did the vials re-cool between heating and measurement or were they measured warm? (did they have the hot plate under them as your measured them?).

Yes, the latter. We will add it to the revised manuscript.
Your total flushing time is somewhat based on flushing volume 'turns'. I noticed on figure one you cite an inlet rate of 35 ml/min. On our 2130 we actually only pull ~25 ml/min. So, I wonder if you have double checked that rate? It might be nice to put a note on line 103 that says something like "time to one full volume can vary Picarro to Picarro".

We will add this in the revised version of the manuscript. As far as we know the picarro does not aim for a certain flow rate per se, it rather aims at keeping the cavity pressure constant (50 torr) by locking the cavity inlet valve to a certain setting and adjusting the cavity outlet valve accordingly. Therefore, the effective flowrate seems to be a function of e.g. ambient air pressure (or whichever pressure is effective/applied at the analyzer's inlet port) or pump performance and will likely be different for different instruments. We have tested 4 different Picarros and found that the flow rates ranged from 27 to 33ml/min, with the mean resulting in ~30ml/min. We will add this information to the revised manuscript, and correct our proposed 35ml/min, which was a typo.

How do the authors identify spurious vials?

We did not exclude any data, and we did not use broken vials. Upon doublechecking whether the vials were properly crimped, we excluded some vials with twistable lids (see above explanation) prior to sampling (i.e. those were never filled or measured). Because we left enough time between measuring two samples, we did not observe any memory effect and therefor did not need to exclude data.

If we theoretically had encountered readings that did not at all fit the expected isotope signature, we would first have checked for the water vapor concentration to see whether we had missed a badly crimped vial/vial with a defective lid/vial with cracked glass. If

that had not been the case, we would have doublechecked in the protocol whether there had been a sampling error/any unusual occurrence during sampling. We will add a brief paragraph on such problems to the revised manuscript to give guidance to the reader.

**Storage time correction**: I think more explanation of your choice to use a generalized storage correction is needed here. From what I can tell from your data, the offset between ambient air and the measured isotopes should dictate how much it moves. For example, the storage correction for d18O from the 'light' isotopes is very different than the one predicted by the 'crazy' heavy. It's relatively easy to imagine a scenario where the ambient air isotopes are very similar to those sampled for an experiment and so just using a light or medium correction would be more appropriate. If the scale of correction is indeed not very different across isotopes, it would be helpful to demonstrate that some way in the supplement.

Storage time correction relates the change between the samples measured on the sampling day (i.e. "0-day samples) to samples measured on subsequent days (i.e. 1,3,4,7,14). For the oxygen isotopes this is only regarding natural abundance ranges as we did not label oxygen. The different isotopic signatures are not considered here since we relate the isotopic change to the storage time. We assumed this change to be similar for each natural abundance isotopic composition. We will give a table in the appendix including the slope and intercept for this relation for each single isotopic composition in the revised version.

These papers are really hard to do, and I applaud the authors for the effort. But given that they are going to apply a linear regression to correct data in the future, I question whether or not they have gotten enough data to truly say that they are representing real variability. Some further discussion of sample size, as it relates to creating a correction factor would be helpful. Further, do you think each lab should create their own correction line or do you think that this is more universally applicable?

We would argue that we provided a large enough data set (i.e. 5 standards *10 replicates per standard =50 data points per storage time) to provide an estimate of the storage effect for our lab conditions.

For this method to be sufficient under the premises that each picarro/each lab uses their own "storage effect time regression" each time they are measuring a large set of samples. This would be the minimum requirement from our point of view for our objective.

Generally, the approximate size of the storage effect would be expected to be similar between labs, however, to obtain an exact storage effect on the respective instruments, we do recommend providing their own correction of the storage effect. Our values can be used as a guideline.

We will add this to the section 4.3 of the revised manuscript.

I appreciated that the authors included a preceding works section – it demonstrated the motivation for their work and helped show context. I also appreciated the discussion of how current system constraints have introduced location and social biasing into the scientific literature.

Thanks ☺

**Technical Corrections:** In this section, I have labeled my correction by line

32-33: I think it would be appropriate to significantly expand this citation list to showcase the variety of kinds of in situ work that is being done. For example, it would also be appropriate to cite Maria Quade's, or T.H.M Volkmann's work here. Beyer et al., 2020 (HESS) would also be a nice addition here.

We will expand the list in the revised manuscript

43: I think it's also fair to cite Orlowski et al., 2016 here

We will add this reference to the revised manuscript

48: Beyer et al., (2020) *HESS* would also be good to cite here

Will be added in the revised version

49: The word choice interferences doesn't sit well with me, I wonder if this sentence could be reworded to make your meaning clearer. Perhaps ".... Direct equilibration between liquid and vapor water in the soil ..."

The word we chose was "inference" not "interference". Does this change the way you feel about the wording? Technically, this is what an in-situ approach does, so we feel the choice of word is appropriate in the context.

91: Were you able to dry the PTFE tubing that goes between the sample vials and the CRDS to eliminate any memory effect from that part of the system? I imagine that could be done quite simply by just having a 'dry' vial that you flush through between samples.

See above answers related to PTFE stickiness and measuring dry vials.

Figure 1: These kinds of figures are very challenging to make well. I appreciate that the photo demonstrates practical complexity in the lab setting. I think this figure could be improved with the addition of a small, simplified cartoon to the side showing all the components. This would help readers hone in on the important components without getting too distracted by all of the real-world lab complexity. Or, another way to make the figure more readable to be to add a small white box behind the text boxes, I had a hard time with the red text in particular.

Thanks, we will use your suggestions and edit the figure accordingly in the revised manuscript.

103: You cite a 35 ml /min pull rate from the Picarro, with a 50 ml container & 10 minutes of flushing that should only be 7 turns (35*10/50 = 7). I'm not sure that nit-pickiness really matters for the scope of this experiment given that the authors observed signal stabilization. But, I think given that this is a methods development paper its most helpful to the community to be hyper-specific about some of these details.

See above answer regarding the picarro flow rate and our calculation. As mentioned, the typo 35ml/min will be changed to 30. However, in the line you refer to here, we cannot see any such calculation. When calculating with 30ml/min suction rate, we get 6 turns. We will add this to the revised version to clarify that the sample will be exchanged several times.

156: A huge advantage of this system over the SWISS is the size and therefore ease of transport (e.g. 50 ml vials vs. 650 ml flasks), so I think one selling point that could be an estimate of total size & weight (just as the authors did with the battery). The SWISS

also requires quite a bit of time consuming construction and plumbing and so some sentence to that effect, and an advantage of this system is that it is easy to set up.

We will add an estimate of weight and size for 100 crimped vials in the revised manuscript. It is 2.5 kg weight and 30x13x50cm is the packaging when re-using the original vial box.

180: Is your data reduction code widely available (e.g. github)? This development paper would be even more helpful to the community if we can also see the data handling process.

The code is available upon request and will be added to a repository once it's completely cleaned.

Figure 3:

I'm not sure if it's the file the authors provided, or a formatting issue from HESS, but it would be great if figure 3 was the full width of the page. If it is an author-side issue, using ggsave you can set the figure width to 6.5 inches - ggsave(plot, "plot.pdf", width = 6.5, units = c("in")).

We will provide a figure with these dimensions for the revision.

Where does ambient air sit in isotopic space relative to the standards measured? I think for the "crazy heavy" water it's easy to see that its trending towards room values, but it'd be nice to have a sense for how far it got towards that value.

We will provide a figure where we can show that at least for the oxygen isotopes a drift towards ambient air is not the case. For the heavily enriched waters this would be easy to assume as it trends toward depletion, but the atmosphere lies within the natural abundance range of the measured standards, and it is there we can see that also for the hydrogen isotopes the drift is not towards ambient air. Which we see, in addition to the dry air samples, as an argument that if there was diffusive exchange with the atmosphere it was a one way process out of the vial into the atmosphere but not vice versa.

190: Please expand on why a wilcox test is appropriate here, and what you hope to learn through it. Unfortunately, many people reading your paper might be unfamiliar with that statistical test – and a short 1 -2 sentence explanation of its use and limitations would help your reader assess suitability.

A Wilcoxon (the typo will be corrected in the revised manuscript) test is a non-parametric approach to detect differences between two groups of data. We chose it to be able to compare the data since they were not normally distributed which would be a requirement for using a parametric test (students t-test). We will add this information to the revised version of the manuscript.

220 – this paragraph as written is a little confusing. I think this could be solved with just a quick additional introduction sentence that says something along the lines of "We observe two different patterns between hydrogen and oxygen isotopes, we first address storage effect on hydrogen isotopes and then oxygen isotopes."

We will add this to the revised version of the manuscript.

243: Ahh, very clever. I initially didn't like that choice, but with the explanation, it makes sense.

Thanks ☺

336 – My feeling is that this contradicts what was stated in the results

We can rephrase this sentence in case this makes it clearer to: "This is the best supported scenario, as the data supports that there is no diffusion…" We want to emphasize that the exact physical process behind this remains unsolved beyond the arguments we deliver in our discussion.

---

## Author Comment (AC3)

**Comment on hess-2022-37**

Anonymous Referee #2

The work presented in HESSD by Magh and colleagues describes a new approach to sample xylem water for isotopic analysis in trees. In this technical note, the authors conducted a lab trial and a field trial to show the suitability of the proposed method. The general motivation of the study is to provide a cost-efficient approach to overcome the limitations of tree water sampling in field conditions and the potential bias of cryogenic extraction. This is important as this is a current point of debate in the literature (i.e. cryogenic limitation) and the proposed method is interesting.

We thank the anonymous reviewer for their extensive and insightful review of our manuscript. We hope to be able to address all their comments and questions sufficiently and provide a comprehensive revised manuscript.

While the study has a great design and is innovative, and the method provides reliable results in the lab, my main concern is that the proposed approach does not seem to provide reliable data from trees in the field (Fig. 7). This is a holdback, as this should be reliable if the goal is to obtain stem water and overcome limitations. It would be important to present more data in the field, i.e., more than one day, and possibly, in more than one tree (since the approach is "easy to use and cost-effective"), and different environmental conditions, i.e. wet vs dry.

As we indicated in the original draft this method originates from an extensive data set obtained in the field from daily observations of 120 trees over the course of 2 months (see "previous attempts" section in the original draft). We developed the lab trial to develop a method that would allow us to overcome the limitations of the method used in said field trial and tested the new method extensively in the lab. We were able to prove that it does work within the addressed uncertainty range, and then additionally tested it on one tree in the field on one day. We feel the scope of the study, to introduce a new measurement method that can provide basis to overcome spatial limitation of in-situ measurements has been accounted for with the lab trial. We agree, the method needs further testing and is far from perfect and fully investigated. We would like to share it at this point anyways so the community can benefit from our experience and start testing it now.

We will address your concerns in the revised version of the manuscript to provide the basis for improvements of the method.

The authors show a statistically significant difference between the *in situ*, defined in the paper as the "gold standard"/ "true value" and the VSVS data, even after the correction for storage is applied. This difference is evident even for the zero-day storage (Fig. 7) for d18O. While for d2H, there is no difference with some days (e.g. 1 and 7), it did not show a clear pattern. Thus, the reliability of the data is uncertain because the effect seemed random (e.g., why is 0-day results different from the i*n situ* and not 1-day?). This is difficult to understand with a single trial in the field.

We agree there are limits to the field data we provided. However, as mentioned above, we aimed at providing a new approach the community can further develop and improve. The reason for the field data to not match the in-situ observations can be manifold and we will address some reasonable explanations in the revised manuscript.

Suppose one can sample vapour with higher or lower water content (ppmV), as the authors even experienced (L166-168), and as we know the water content of the wood

changes largely depending on the water status (e.g. high water content in the wood because of well-water conditions or low water content due to water stress), how would that interfere with VSVS and storage time, or even the proposed correction? Lab trials usually show fractionation in vapour samples with low ppm. Since the authors are offering/reporting a new approach, it would be interesting to understand this before we, as a community, start to apply the technique broadly. Would it be beneficial to use larger vials during drier periods? Have the author's tested different volumes of vials for larger vapour storage?

Vapor pressure is related to the curvature of the bounding surface and would therefore only be reduced in either very dry soils or very dry tree tissues (see e.g. (Thomson, William Sir 1871). Those conditions in a tree would most likely lead to irreversible cavities and therefor dying of the tree (see e.g. (N. McDowell et al. 2008; N. G. McDowell et al. 2022). So, we would assume there is no physical basis for dry conditions in viable trees to lead to smaller vapor concentrations. We will address this in the revised discussion.

We have initially tested even smaller vials (20ml) and discarded those in the very beginning as the length of measurement did not allow for a stable 2 min isotope reading. We will add this to the previous attempts section in the revised manuscript. Havranek et al. (2020) did provide a similar system to ours, that they did test in the lab, showing that large bottles (700ml) provide more stable storage and isotope readings. However, their bottles are very large, and custom made (which makes their approach more expensive and less flexible).

Regarding the borehole, have the authors monitored the wounding effect? Similar to sap flow systems, where the wounding effect can influence sap flow rate measurements (e.g. Wiedemann et al., 2016; Peters et al., 2018), one would expect that we observe a similar effect when sampling for precise natural abundance where the boreholes are much larger than the small sap flow needle. Can the authors comment on this and discuss this limitation? For example, if multiple samples need to be collected from a forest in high- temporal resolution, for how long can one rely on the same borehole? Additionally, conifer species tend to produce resin near the wound, this could additionally result in spectral contamination. How could one define if spectral contamination is an issue in this system?

We have monitored the pitch/resin production after drilling and cleared out the resin using Acetone (see line 141) every day. Only when the production ceased did we install the fittings (after 4 days see below). Additionally, we checked whether there was new resin coming in after five weeks before connecting the tree to the VSVS (see also in an answer to your specific comments below), and there was not.

To check for spectral contamination (e.g. associated with organics/Voc's) we regularly checked the methane ("CH4") variable recorded simultaneously on the Picarro during the measurements. We compared it to the values when measuring the standards which should have close to no spectral contamination. We did not observe any differences. Further, VOC pollution is rather associated with e.g. the EQ bag method, where prolonged storage time may facilitate VOC accumulation which then potentially flaws the obtained isotope readings.

Regarding the reliance on one borehole over time we have to refer to the already published studies conducted by (Marshall et al. 2020; Beyer, Kühnhammer, and Dubbert 2020; Kühnhammer et al. 2022). The longest time using the same borehole was studied in Kühnhammer et al. 2021. They did observe reliable results over the course of 2.5 months in a

tropical dry forest. However, the authors also call for a more systematic investigation of how long a borehole can be used. We will add this to the revised discussion.

I think this study is missing a direct comparison with the cryogenic system. In the introduction, the authors refer to the potential bias of the method while this is also the "state-of-the-art extraction process" in this field, so how do the results compare with cryo? Or even the direct-equilibrium bag method?

Cryogenic extraction has often been shown to result in heavily biased data and should therefore no longer be considered the state of the art in our opinion. See for examples (Orlowski, Breuer, and Mcdonnell 2016; Allen and Kirchner 2021; Chen et al. 2021). We will change this "state-of-the-art extraction process" statement in the introduction since cryogenic extraction is no longer that.

We respectfully disagree with the comparison to cryogenic extraction. Our aim was the comparison between in situ and „bottled" vapor to show the potential effect of our storing approach. Whether in-situ is "correct" in terms of representing the stem water is not the scope of this study.

Regarding the comparison to the direct equilibrium bag method: the information we would obtain would be one single observation (most likely from the day installing the borehole) and the xylem composition would for sure have changed by the time the "settling period" of the same was over.

We have tried sampling vapor directly into the bags most commonly used for the bag equilibration method, but handling proved to be much harder than with the vials. We will add this information to the revised manuscript to the section "previous attempts".

Specific comments:

Title: The title mentions soil, but no tests or trials are done for soil in this work.

We will revise the title as also mentioned in the answer to the community comment. It will be more concise.

In line 36, the author refers to "matrix-bound" (assuming soil, as the previously mentioned soil matrix in the above paragraph). Still, in the last lines (L40-42) of this paragraph (L36-42), the authors use references that discuss cryogenic bias that relates to plants (Chen et al., 2020 and Allen and Kirchner, 2021). It would be helpful to be clearer and refer specifically to plant cryogenic bias in the text. Or, if the authors want to refer to cryogenic bias in both soil and plants, it would be helpful to mention it more clearly with appropriate references to both cases.

We will revise this so that the references match the text.

Lines 45- 46. What do the authors mean by altering their physiology?

We mean that the water flow will be disrupted when sampling tree cores on the same individual repeatedly to e.g. obtain a comprehensive understanding of water uptake over time. That leads to altered water transport patterns and subsequently alters the physiological functioning. We will elaborate on this in the revised manuscript.

Line 62: Add Kuhnhammer et al., 2021 along with Beyer et al., 2020 here as well. We will provide the suggested reference in the revised version.

Line 81: What was the volume of the standard water (in the larger vial)? Did the authors try different volumes of crimp neck sample vials? Why 50 ml was the selected volume?

The standard bottles were 250ml "Schott" lab bottles containing 50ml of the standard water. We will add this information to the revised manuscript.

As for the size of the crimp vials, see above. We tried 20ml vials first, but ultimately we were unhappy with the sample volume as it was too small to give a sufficiently long isotopic plateau during the measurements.

Line 139: Give the scientific name to the two species.

We will add the Latin names of the species to the revised manuscript.

Line 141: Give an estimate of the "several days" (e.g. ~ 5 days)

We waited 4 days between drilling the borehole and installing the fitting. We will add the information to the revised manuscript.

Line 143: It would be helpful to the reader if you already refer to the schematic figure here (Fig S1), and perhaps bring it to the main body of the paper (nice figure!).

We will add the figure to the main body in the revised manuscript and refer to it at this point.

L144: Maybe state clearly that one of the scots pine previously connected with the *in situ* system was monitored with the new system VSVS.

We will add the word previously to the sentence in the revised version of the manuscript.

L149: Was the borehole flushed again or any treatment used after the five weeks before the change in the system? Did the authors detect any wounding effect in the borehole? Since this is a technical note, these details should be clearer so others can replicate the method.

There was no further treatment of the borehole, except visual inspection regarding pit/resin production (there was none) before we connected it to the VSVS for the first time.

L161: What was the air temperature in the field during the vapour sampling? How does change in air temperature affect the sampling (e.g. from wood to air)? Or if sampling in days with different air temperatures? It would be important to include the first answer here since this is a methods paper and the later ones in the discussion.

Air and source (i.e. borehole) temperature was monitored constantly. However, until entering the crimped vial the PTFE tube was heated using a heating line and insulated. Because we re-heated the vials prior to measuring and during the measurement, we were not concerned about condensation that might occur during the storage period. Therefore, the source temperature defines the conversion between vapor and liquid isotope signatures. We will elaborate on this principle in the revised MM and mention it again in the discussion.

L175: It is not too clear why the in-situ n = 2. How was it determined? Please clarify this part or re-arrange the text to be more explicit.

The in-situ system was set up so that each tree was measured every 4 hours. During the time we sampled for the VSVS we disconnected the tree from the in-situ system and re-connected it to the in-situ system each time the trees measurement time came up in the schedule for the in-situ system. On that day we were able to make 2 observations using the in-situ system within the time we sampled for the VSVS, that equals n=2. We will rearrange the text to explain this in the revised manuscript.

L252-253: It would be helpful to show the atmospheric data in the supplementary information along with the samples.

We will add the required data to the revised manuscript.

L282: It would be helpful to see the data similarly to Figure 6. The raw along with the corrected for the field measurement.

We will adapt figure 7 accordingly in the revised manuscript.

L280-282: Didn't VSVS also fail to return the in-situ when compared with the other days (i.e. 1, 3, 7 and 14) and not only the "0-day"? Perhaps state it more directly.

We will revise this sentence to clearly show that it failed to return the oxygen composition regardless of storage time.

L290-295. "3.3 Time and Cost Efforts" – This is not a result per se but part of the discussion. For a more comprehensive comparison, one should also state what type of cryogenic extraction the authors refer to, as the reference is not enough as this is important to the reader. The time efficiency one should also discuss field-set uptime (e.g. how long does it take to set up the VSVS and in situ in the field?). This would be helpful to understand if short-term studies would still benefit from this approach.

We will move this to the discussion in the revised manuscript. We refer to using an extraction line following the Königer approach (Koeniger et al. 2011). The time effort includes the system setup, which takes longer for the in-situ than for the VSVS, since the latter only requires the fittings being installed, while the tubing is movable from tree to tree. It then takes longer to measure the samples, where the in-situ system does not require any additional work. This is how we ended up with the same amount of time needed for in-situ and VSVS. We will add more information to the table in the revised version.

L308-309: This is a bit of an overstatement for the field conditions. The VSVS proposed method results were not statistically similar to the defined "gold standard"/"true value" (*in situ*) for d18O and were somewhat similar for the d2H.

We will revise to something like: „sufficiently well in the lab and in the field study regarding the tracer detection in the hydrogen isotopes. We conclude with a call for more field data for this method to obtain field approval also for delta18O."

Fig. 6 Very minor comment: It would be nice to align the x-axis between the two plots.

We will revise the figure so that the axes align

Fig. 7 Very minor comment: Makes the asterisks larger; it is difficult to see them.

We will revise the figure so that the asterisks are larger.

References:

Wiedemann, A., Marañón-jiménez, S., Rebmann, C., Herbst, M., & Cuntz, M. (2016). An empirical study of the wound effect on sap flux density measured with thermal dissipation probes. Tree Physiology, 36, 1471–1484. https://doi.org/10.1093/treephys/tpw071

Peters, R. L., Fonti, P., Frank, D. C., Poyatos, R., Pappas, C., Kahmen, A., Carraro, V., Prendin, A. L., Schneider, L., Baltzer, J. L., Baron-Gafford, G. A., Dietrich, L., Heinrich, I., Minor, R. L., Sonnentag, O., Matheny, A. M., Wightman, M. G., & Steppe, K. (2018). Quantification of uncertainties in conifer sap flow measured with the thermal dissipation method. New Phytologist, 219, 1283–1299. https://doi.org/10.1111/nph.15241

References used in the answers to the reviewer:

Allen, Scott, and James Kirchner. 2021. 'Potential Effects of Cryogenic Extraction Biases on Inferences Drawn from Xylem Water Deuterium Isotope Ratios: Case Studies Using Stable Isotopes to Infer Plant Water Sources'. *Hydrology and Earth System Sciences Discussions*, no. January: 1–15. https://doi.org/10.5194/hess-2020-683.

Beyer, Matthias, Kathrin Kühnhammer, and Maren Dubbert. 2020. 'In Situ Measurements of Soil and Plant Water Isotopes: A Review of Approaches, Practical Considerations and a Vision for the Future'. *Hydrology and Earth System Sciences* 24 (9): 4413–40. https://doi.org/10.5194/hess-24-4413-2020.

Chen, Yongle, Brent R. Helliker, Xianhui Tang, Fang Li, Youping Zhou, and Xin Song. 2021. 'Stem Water Cryogenic Extraction Biases Estimation in Deuterium Isotope Composition of Plant Source Water'. *Proceedings of the National Academy of Sciences of the United States of America* 117 (52): 33345–50. https://doi.org/10.1073/PNAS.2014422117.

Havranek, Rachel E., Kathryn E. Snell, Brett Davidheiser-Kroll, Gabriel J. Bowen, and Bruce Vaughn. 2020. 'The Soil Water Isotope Storage System (SWISS): An Integrated Soil Water Vapor Sampling and Multiport Storage System for Stable Isotope Geochemistry'. *Rapid Communications in Mass Spectrometry* 34 (12): 1–11. https://doi.org/10.1002/rcm.8783.

Koeniger, Paul, John D. Marshall, Timothy Link, and Andreas Mulch. 2011. 'An Inexpensive, Fast, and Reliable Method for Vacuum Extraction of Soil and Plant Water for Stable Isotope Analyses by Mass Spectrometry'. *Rapid Communications in Mass Spectrometry* 25 (20): 3041–48. https://doi.org/10.1002/rcm.5198.

Kühnhammer, Kathrin, Adrian Dahlmann, Alberto Iraheta, Malkin Gerchow, Christian Birkel, John D. Marshall, and Matthias Beyer. 2022. 'Continuous in Situ Measurements of Water Stable Isotopes in Soils, Tree Trunk and Root Xylem: Field Approval'. *Rapid Communications in Mass Spectrometry* 36 (5): e9232. https://doi.org/10.1002/rcm.9232.

Marshall, John D., Matthias Cuntz, Matthias Beyer, Maren Dubbert, and Kathrin Kuehnhammer. 2020. 'Borehole Equilibration: Testing a New Method to Monitor the Isotopic Composition of Tree Xylem Water in Situ'. *Frontiers in Plant Science* 11 (April): 1–14. https://doi.org/10.3389/fpls.2020.00358.

McDowell, Nate G., Gerard Sapes, Alexandria Pivovaroff, Henry D. Adams, Craig D. Allen, William R. L. Anderegg, Matthias Arend, et al. 2022. 'Mechanisms of Woody-Plant

Mortality under Rising Drought, CO2 and Vapour Pressure Deficit'. *Nature Reviews Earth & Environment*, March. https://doi.org/10.1038/s43017-022-00272-1.

McDowell, Nate, William T Pockman, Craig D Allen, David D Breshears, Neil Cobb, Thomas Kolb, Jennifer Plaut, et al. 2008. 'Mechanisms of Plant Survival and Mortality during Drought: Why Do Some Plants Survive While Others Succumb to Drought?' *New Phytologist* 178 (4): 719–39. https://doi.org/10.1111/j.1469-8137.2008.02436.x.

Orlowski, Natalie, Lutz Breuer, and Jeffrey J. Mcdonnell. 2016. 'Critical Issues with Cryogenic Extraction of Soil Water for Stable Isotope Analysis'. *Ecohydrology* 9 (1): 3–10. https://doi.org/10.1002/eco.1722.

Thomson, William Sir. 1871. 'On the Equilibrium of Vapour at a Curved Surface of Liquid'. *The London, Edinburgh, and Dublin Philosophical Magazine and Journal of Science* 42 (282). https://doi.org/10.1080/14786447108640606.